# Perception, Memory, and Inference:
# The Trinity of Machine Learning

**Adam Dahlgren Lindström**[*]  and  **Johanna Björklund**[†] ,  **Frank Drewes**

Department of Computing Science, Umeå University

{dali,johanna,drewes}@cs.umu.se

## 1   Introduction and background

As an answer to recent contributions about the conjectured impossibility of learning meaning from surface form alone, and the dangers of large language models, we argue in this paper that an explicit distinction should be made between (i) perception, (ii) (explicit) memory, and (iii) inference. We envision a triad of interacting subsystems with corresponding responsibilities (see Figure 1). *Perception* provides the interface between the system and its environment by learning and recognising patterns, which is typically realised in the form of a language model. Explicit *memory* is a structure of concepts and relations between the concepts, in other words, a knowledge base of facts. *Inference*, finally, corresponds to mathematical or rule-based reasoning and provides, for example, classical logic and arithmetic. We note here that such axiomatic systems cannot be deduced from data, only conjectured and tried against data. Our position is that natural language systems should thus combine continuously updated language and other perception models (e.g., computer vision) with one or more symbolic knowledge bases that relieve the models from learning concepts and their relations, and finally one or more inference engines to provide formal reasoning. In a multimodal natural language context, deep learning is a powerful and versatile tool for processing text and images as far as perception goes. Bender and Koller conjecture that one cannot learn the relation between the surface form of language and its communicative intent from the former alone [Bender and Koller, 2020], and Bender et al. [Bender et al., 2021] argue that the race for bigger and bigger datasets and language models should be seriously questioned as it carries considerable environmental, social, and scientific risks. Training these models consumes huge amounts of energy, big datasets are skewed against minorities and the underprivileged, and the focus on size hampers progress towards more intelligent solutions. We think that these are valid points, but would like to qualify the by two comments:

First, learning from large datasets is not inherently problematic. A human who is given access to the type of datasets used for knowledge extraction can, over time, be expected to learn useful facts and be wiser for the experience. For example, a reader who has been taught how the shape of the earth can be derived from physical observations will not soon adopt the flat earth theory from reading about it in online media. Human rationality makes us more robust than neural networks with respect to how new data points affect us. Our beliefs, values, and reasoning abilities are the "knowledge base" that makes us, if not immune, then much less susceptible to integrating misleading information. In contrast, current deep learning does not have the capability to discard a data point as false and choose not to learn from it, but must take everything encountered at face value and adjust the network parameters accordingly. What makes large datasets problematic are therefore the specific conditions of the learning process that characterise deep learning. We believe that the addition of symbolic knowledge would make overly large models unnecessary: having access to a multiplication algorithm is more space efficient than memorising a huge multiplication table, which can in any case never be complete.

Second, learning surface form is a challenging research problem and the improvement of existing techniques can create substantial value if, as argued earlier, language models are not viewed as complete world models, but rather as models of form that may be compatible with any number of worlds. In their work, Bender and Koller do not address the downstream systems that make use of pretrained language models. However, we would like to suggest that it is precisely at the point of application that the language model can be said to infer meaning, namely by bridging the gap between the perceived world and the system's internal representation of the world.

## 2   Position statement

Let us stop and reflect on how humans learn. Comparing humankind to other species, our success is largely due to collective learning: we systematically codify knowledge so that we can store and transfer it in a compact form, relieving individuals from having to learn everything from scratch. If we look farther than our predecessors, it is because we are standing on their shoulders. We teach pupils simple algorithms to multiply numbers rather than expecting them to figure it out themselves, or to memorise each product of two numbers as a separate fact. We also teach them how to read, in order to update themselves with new facts from newspapers or the Internet. To mimic this to a certain degree, and thus overcome

---

[*]Contact Author

[†]The authors are given in alphabetical order. This work is partially funded by the Swedish Research Council, Grant number 2020-03852, and the Wallenberg Autonomous Systems and AI program.

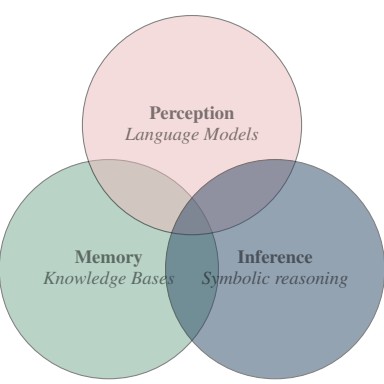

Figure 1: The triad of capabilities and components advocated.

deficiencies such as those listed in [Bender and Koller, 2020; Bender et al., 2021], we propose that machine learning systems in general and language learning systems in particular should be equipped with similar abilities. In support of this proposal, we would like to point out:

1. Merely using more data seems to us an, if not outright impossible, then at least an inefficient, means of producing systems with human-like faculties of reasoning.

2. In contrast to, e.g., recent works that attempt to use word embeddings as knowledge bases [Petroni et al., 2019; Bouraoui et al., 2020], we believe it is more effective to realise the faculties of perception, reasoning, and memory as separate computational unities. In practice, this means integrating neural networks with external knowledge bases and inference engines.

3. The ability to assess new data points in the light of accepted knowledge and, if appropriate, disregard them rather than incorporating them into the model, can make systems more robust against the imperfections of training data, and make the learned model more coherent.

## 3 Related work

There is a rapidly growing body of literature on hybrid machine-learning systems (see e.g. [Wang et al., 2019; Hohenecker and Lukasiewicz, 2020a; van Bekkum et al., 2021]). We discern two main lines of work. The first combines perception and inference, in other words, deep learning and rule-based or neuro-symbolical reasoning. A good example is AlphaGo [Silver et al., 2017] which augments Monte Carlo search trees with deep neural networks. An example from computer vision is DeepProbLog [Manhaeve et al., 2018] which separates perception from inference. More precisely, it employs a simple convolutional neural network to identify digits from MNIST, supported by probabilistic logic programming for modelling and reasoning.

A survey of perception-inference hybrid systems is given by [Raedt et al., 2020] which address neuro-symbolical and statistical relational approaches to integrating learning and reasoning. The authors provide examples that leverage the strength of both methods, such as [Ellis et al., 2018]. They also identify open challenges, e.g., leveraging the effectiveness of deep learning for symbolical representation learning.

The second line of work combines perception and explicit memory, that is, deep learning and knowledge bases. Lecue investigates the role knowledge graphs have in explainable AI [Lecue, 2019] and explains how knowledge graphs can be integrated with deep neural networks to aid explainability, to bootstrap natural language models, and to disambiguate between word senses when uncertainty arises. Knowledge graphs are also studied in the context of neural network architectures, see for example the recent review of graph neural networks [Zhou et al., 2020]. Reasoning over knowledge bases is explored in e.g. [Minervini et al., 2020; Hohenecker and Lukasiewicz, 2020b; Qu et al., 2021], and is reviewed in [Chen et al., 2020].

Another set of writings treat knowledge bases in a multi-modal framework. Multimodality generally means that information is drawn from a heterogenous source of data, the most studied combination being language and visual data. Here, knowledge bases help connect the modalities [Pezeshkpour et al., 2018; Krishna et al., 2016; Zhu et al., 2015; Kannan et al., 2020]. For purely visual data, it is known that knowledge graphs can aid machine vision tasks [Marino et al., 2017], and also that unimodal language models stand to benefit from the addition of knowledge bases [Petroni et al., 2019]. How other modalities and how the construction of data sets limits what a model can learn is outlined in [Bisk et al., 2020].

Covariate shift [Sugiyama and Kawanabe, 2012; Sugiyama et al., 2007] and concept drift [Gama et al., 2014; Lu et al., 2018] are related to the problem of disregarding data points using existing knowledge. A difference between training and test distributions is a covariate shift, and [Schneider et al., 2020] improves model robustness using covariate shift adaptation. Shifting over, e.g., time is called concept drift, and [Webb et al., 2016] characterises such drifts.

## 4 Key challenges

Although a more diverse set of tools can be expected to have advantages as opposed to a total reliance on deep learning or neuro-symbolic methods, it also leads to new challenges:

- Which overall system architecture is needed to enable a seamless integration of perception, memory, and inference subsystems as indicated in Figure 1?

- Can we design learning strategies that validate new facts using the knowledge base and inference system, and integrate new data into the model only if it is consistent with accepted knowledge and values?

- Can the language model and the knowledge base evolve over time, and how do we keep them aligned as language changes and the meaning of words start to drift?

- Can we integrate the inference engine and the language model to enable heuristic search and inference that make use of the capabilities of the language model?

Given the richness and diversity of solutions that can be attained by fusing perception, explicit memory, and inference, we believe that the fields of machine learning and neuro-symbolic inference can draw on the strengths of each other to gain a truer understanding of meaning.

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
