# OpenReview forum: "Perception, Memory, and Inference: The Trinity of Machine Learning"
_ijcai.org/IJCAI/2021/Workshop/NSNLI — NSNLI Oral_

### Official Review · Reviewer_SJyc · 2021-05-23
**Raises some reasonable points, but needs more thinking and justification.**

**Rating:** 3
**Confidence:** 4

**Review:**

This is a position paper about what directions for future work in neurosymbolic learning methods. There are three main parts to the position:

1. Language models will not reach human-like reasoning from an increase in the size of the training set alone.
2. (paraphrasing) Learning systems should have explicit modules for perception, inference, and memory.
3. Machine learning methods need the ability to disregard data points.

These are all reasonable positions to take, but the abstract does not seem to provide evidence in support of these propositions, or new thinking about their implications. Many of the positions are supported by introspection on human reasoning and educational systems; this is problematic. More specifically:

Conclusion 1: "it cannot be expected that only increasing the size of the data" Many researchers would agree with this statement, but I am missing the evidence for it. Many people have been surprised at the increasing capabilities of every larger models. Why might they not be surprised again?

Conclusion 2: "It is meaningful to make a distinction..." This is vague, and as far I understand it seems noncontroversial. Is there anyone who would disagree that perception, memory, and inference are different tasks? If you mean to make a stronger claim than that, it should be made more explicitly. "integrating neural networks with knowledge bases and inference engines": indeed, that is an interesting research area, and the paper cites some work in this area. How does the analysis that you are doing provide insight into these areas, except that we should do more of it?

Conclusion 3: This is a potentially fruitful area of research, and indeed it has been explored in research on handling covariate shift and concept drift.

In general, I would argue that we should be skeptical about statements about human learning like those in the first paragraph of Section 2. The way that people learn might not be the most effective way to make intelligent systems. And many of the most important and challenging capacities of animal intelligence --- such as perception and memory! --- are not often taught in school or in books.

For this reason, I unfortunately cannot recommend including this work in the workshop program, but I would certainly encourage the authors to participate.

---

### Decision · Program_Chairs · 2021-05-27

**Decision:**

Accept (Oral)

**Comment:**

Despite the negative review, we have decided to include the paper in the workshop's program.
In the end, the goal of the workshop is to give the participants a chance to present their work and get useful feedback
The provided review already provides very useful feedback, and we hope the participation at the workshop will provide even more.